# Population-Based Linkage of Big Data in Dental Research

**DOI:** 10.3390/ijerph15112357

**Published:** 2018-10-25

**Authors:** Tim Joda, Tuomas Waltimo, Christiane Pauli-Magnus, Nicole Probst-Hensch, Nicola U. Zitzmann

**Affiliations:** 1Department of Reconstructive Dentistry, University Center for Dental Medicine Basel, 4056 Basel, Switzerland; n.zitzmann@unibas.ch; 2Department of Oral Health & Medicine Dentistry, University Center for Dental Medicine Basel, 4056 Basel, Switzerland; tuomas.waltimo@unibas.ch; 3Department of Clinical Research & Clinical Trial Unit, Faculty of Medicine, University of Basel, 4031 Basel, Switzerland; christiane.pauli-magnus@usb.ch; 4Department of Epidemiology & Public Health, Swiss Tropical & Public Health Institute Basel, University of Basel, 4051 Basel, Switzerland; nicole.probst@unibas.ch

**Keywords:** big data, patient-generated health data (PGHD), register-based controlled (clinical) trials [RC(C)T], epidemiological research, public health

## Abstract

Population-based linkage of patient-level information opens new strategies for dental research to identify unknown correlations of diseases, prognostic factors, novel treatment concepts and evaluate healthcare systems. As clinical trials have become more complex and inefficient, register-based controlled (clinical) trials (RC(C)T) are a promising approach in dental research. RC(C)Ts provide comprehensive information on hard-to-reach populations, allow observations with minimal loss to follow-up, but require large sample sizes with generating high level of external validity. Collecting data is only valuable if this is done systematically according to harmonized and inter-linkable standards involving a universally accepted general patient consent. Secure data anonymization is crucial, but potential re-identification of individuals poses several challenges. Population-based linkage of big data is a game changer for epidemiological surveys in Public Health and will play a predominant role in future dental research by influencing healthcare services, research, education, biotechnology, insurance, social policy and governmental affairs.

## 1. Introduction

Big data and its impact on society is an omnipresent topic affecting all spheres of our lives. We are now more globally connected than ever, and society continuously advances into data-driven infrastructures and organizations. The ‘digital revolution’ has the potential to enrich society by informing social policy decisions, but might also cause harm and put sensitive information at risk of abuse. How to best harness and use big data for social good remains a core focus within the biomedical community [1].

Population-based linkage of patient-level information is a powerful tool in health economics. Health data can be gathered from routine care process and other expanded sources including data on the social determinants of health, such as material circumstances (e.g., housing, transport, food security/safety, environmental degradation). Big data collaborations involve a diverse range of stakeholders with different analytical, technical and political capabilities (Figure 1). 

The linkage of individual patient data obtained from different sources opens new strategies for research. Establishing large population-based patient cohorts may help identify unknown correlations of symptoms and diseases, novel prognostic factors, new treatment concepts, new revenue sources and facilitate the evaluation of the healthcare system [2]. The linkage of patient-level information to population-based citizen cohorts and biobanks provides the required reference of diagnostic and screening cutoffs that could detect new biomarkers through personalized health research [3]. 

## 2. Data Collection and Processing 

As conventional methods for collecting data such as (prospective) clinical trials have become more complex due to high costs, time expenditures and difficult patient recruitment, linked individual-level data is a valuable and promising additional tool in healthcare related science [4]. The strengths of register-based controlled (clinical) trials (RC(C)T) is that these are well-characterized, particularly for medical and dental research, provide comprehensive information on hard-to-reach populations and allow observations with minimal loss to follow-up, but require large sample sizes. This type of trial generates evidence with a high level of external validity [5]. 

Collecting data is only valuable if this is done systematically according to harmonized and inter-linkable data standards to produce high-quality data. This will avoid the garbage-in-garbage-out issue that has afflicted big data’s reputation in the past. The ubiquitous generation and storage of data inevitably leads to huge accumulation of information. Efficient data management not only involves the assessment and storage of information, but must also ensure that the data can be accessed quickly utilizing user-friendly software for fast and easy filtering [6]. However, the best data processing and analysis algorithms are only as good as the reliability and validity of the original data. Despite best efforts, inadequacies in data quality in general do occur. A crucial problem is missing and incomplete data, which can occur when data is not captured in a standardized manner or not recorded for specific population cohorts [7]. 

Digital information that is not yet being used is known as ‘dark data’. Do we as biomedical researchers have the obligation to exploit the wealth of dark data, which may contain vital information that could lead to future achievements such as cures for diseases? However, these data may be of low quality, records may be incomplete, processed incorrectly, or stored in file formats or on devices that have become obsolete over time. Furthermore, by the time the data can be cleaned and processed it may be too old to be useful and would distract us from other discovery research, given that we live in a resource constrained environment.

## 3. Protection of Human Rights

The use of linked biomedical data to support register-based research poses a unique set of methodological challenges, such as disclosing sensitive information about individuals whose consent cannot be easily obtained in retrospect. Data anonymization is a type of information sanitization whose primary intent is privacy protection. It is the process of either encoding or removing personally identifiable information from datasets. Anonymization methods include encryption, hashing, generalization, pseudonymization and perturbation. De-anonymization is the reverse engineering process used to detect the source data. The most common technique of de-anonymization is cross-referencing data from multiple sources [8]. The potential for re-identifying individual patients from anonymized data poses unique challenges to biomedical research, as the protection of patients’ privacy is paramount [9]. Several challenges remain unresolved with respect to the use of anonymized patient data for research purposes: appropriate security procedures (including establishing access permissions to data identifiers) and novel algorithms for statistical analyses and interpretation of generated data need to be developed and implemented. A generally accepted code of conduct has to be defined and established for the ethical and meaningful use of register-based patient data for an optimum impact [10]. 

## 4. Implications for the Healthcare System

The successful establishment of population-based research is significantly dependent on national legal law regulations and the specific healthcare system. Countries with government-driven healthcare systems have the advantage over those organized in a private insurance network [11]. Consistent diagnostic terminology will greatly facilitate the collection of uniform data and subsequent data pooling and standardized diagnostic terminologies are routinely used in several dental disciplines at the local (hospital), state, national or even continent level [12]. Similar to existing projects in human medicine, such as the Human Genome Project (www.genome.gov) or the Research Collaboratory for Structural Bioinformatics Protein Data Bank (www.rcsb.org), the goal for using linked population-based data in dentistry has to be the systematic collection and secure management of coded individual patient information gathered from dental providers. Only by implementing standardized protocols can high-quality data be obtained that enables comprehensive analyses, critical interpretation and clinically useful transferable conclusions [13]. This implies, however, a universally accepted ‘general patient consent’. At the same time, ethical concerns arise questioning the protection of the personal rights [14]. Again, the unsolved issues are anonymization of the patient and the potential risk of publication of identifying keys. Furthermore, if a patient subsequently retracts their consent, what happens to already registered data? 

Moreover, the technological infrastructure has to deal with a continuously growing amount of data that might outstrip the capabilities of data analysis. Highly specialized biomedical informatics researchers are needed and new disciplines and job descriptions may be required in the fields of developmental software solutions, statistical calculations, future epidemiological model simulation and healthcare analytics [15]. 

The potential influence and impact of linked individual-level data on national healthcare systems involving the insurance industry, as well as on social policy and law, cannot be anticipated today. Technological progress in health information technology ecosystems will reveal the diversity of possibilities in the future. Of paramount importance is the responsible management of patient-level information considering ethical issues [16]. 

## 5. Role of Academia

Academic dental institutions and large dental service providers are organized in multi-departmental structures, usually Prosthodontics, Restorative Dentistry, Periodontology, Oral Surgery and Orthodontics. Instead of exploiting the potential synergistic benefits of organizing this umbrella-like infrastructure into academic clusters, individual dental disciplines tend to work in isolation. This results in redundant and non-standardized diagnostics within the different departments, delayed patient treatment due to in-house referrals and increased administration. A translationally integrative concept with standardized dental diagnostic protocols could enhance patient flows and reduced overall treatment time and support interdisciplinary linked-therapy planning with improved quality, higher efficiency and increased patient satisfaction [17]. Cluster-wise data collection will change the dental research environment and reveal unseen opportunities. Linking standardized dental diagnostics with biomedical patient-level data registering will provide the information needed to better understand the epidemiology and etio-pathogenic pathways of oral diseases and may identify unmet needs in dental science. Multi-centric and interdisciplinary large-scale RC(C)Ts will advance new options and reveal comprehensive results in an unprecedented way. 

## 6. Role of Private Professionals

The gathered information of population-based RC(C)Ts can provide an understanding of current and future applications of personalized dental medicine. Efficient and more rapid adoption of personalized dental research will help to improve prevention and rehabilitation concepts of oral diseases. Here, the patient and the group of diverse healthcare providers such as private professionals as well as academic dental institutions and large dental service providers will benefit equally from this development. In the future, private dental professionals will increasingly generate digital data in collaboration with supraregional science centers. Established practice networks will help to extend the overall data pool besides academic dental institutions. A major challenge is the compliance with quality standards in data acquisition, storage and safe transfer. This will have an impact on the daily routine for the general dental practitioner [17]. 

## 7. Conclusions

Population-based linkage of big data is a game changer and will play a predominant role in future dental research by influencing healthcare services, research, education, biotech and pharma industry, insurance business, social policy, governmental affairs and legal law regulations. Further developments in information technology such as artificial intelligence and machine learning will even accelerate data acquisition and analysis. 

Major benefits of population-based linkage of dental data with standardized diagnostic terminology include:
Systematic filtering of patients for prospective research applying trial-specific inclusion criteria;High quality clinical research with large-scale sample sizes and accelerated recruitment;Immediate access to clustered retrospective data;Scientific evidence with low bias and high levels of external validity;Simplified translation from laboratory to clinical science;Epidemiological surveys for Public Health related statistics; andEarly identification of future research needs.


## Figures and Tables

**Figure 1 ijerph-15-02357-f001:**
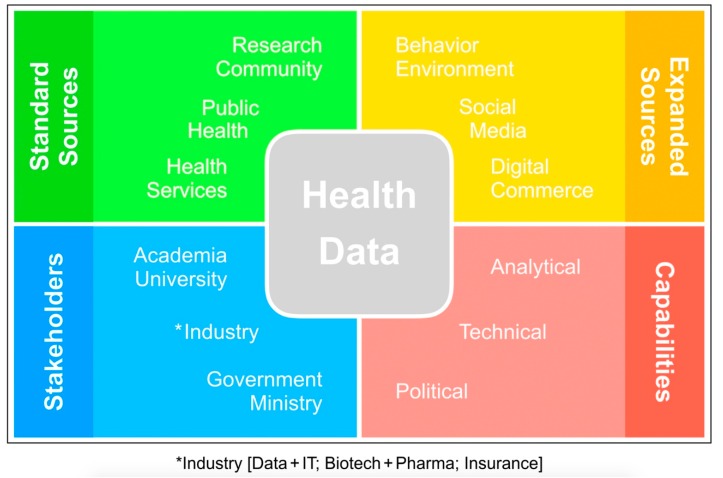
Health data sources, associated stakeholders and capabilities.

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
