# Peer review of "Population-Based Linkage of Big Data in Dental Research"

_ijerph, 2018, doi:10.3390/ijerph15112357_

Round 1

Reviewer 1 Report

very interesting narrative review on a novel topic in digital dentistry, the implications of big data are yet to be explored and this paper already gives an excellent overview on the topic. i suggest to accept this paper in its present form. 

Author Response

Thank you very much for reviewing our submitted manuscript. 

Reviewer 2 Report

the present paper deals with the impact of big data in dentistry. i would like the authors to shortly expand this manuscript in order to better focus on the impact of big data on the private professionals (dentists) because there are i guess several implications but since the topic is new it is quite difficult to identify them. recently, all UE countries have adopted new legislation for data protection and this is having an impact on the common dentists work. i would like to know what the authors think about this and also i would like to know if other implications (not only related to data privacy) should be considered. for example everyday on the social networks (Facebook, etc) colleagues from all over the world post pictures of patients, not only intraoral - clinical pictures but also pictures of the smile and the face, and i am not sure these colleagues have obtained authorization for this. it looks to me very difficult to control this world that is spreading incredibly and i would like the authors to discuss their point of view. what are the limits? what are the further advantages of big data? i was wondering about big data and guided implant surgery. is there any possible implication? and with regard to machine learning? anyway i congratulate with the authors for their brillant work. 

Author Response

Thank you very much for the suggestions and valuable comments.

“The present paper deals with the impact of big data in dentistry. I would like the authors to shortly expand this manuscript in order to better focus on the impact of big data on the private professionals (dentists)because there are I guess several implications but since the topic is new it is quite difficult to identify them. Recently, all EU countries have adopted new legislation for data protection and this is having an impact on the common dentists work. I would like to know what the authors think about this and also I would like to know if other implications (not only related to data privacy) should be considered. For example everyday on the social networks(Facebook, etc.) colleagues from all over the world post pictures of patients, not only intraoral - clinical pictures but also pictures of the smile and the face, and I am not sure these colleagues have obtained authorization for this. It looks to me very difficult to control this world that is spreading incredibly and I would like the authors to discuss their point of view. What are the limits? What are the further advantages of big data? I was wondering about big data and guided implant surgery. Is there any possible implication… and with regard to machine learning? Anyway I congratulate with the authors for their brilliant work.”

What is the impact of big data on the private professionals (dentists)?

We absolutely agree that it is a very interesting and important issue. Following the reviewer’s suggestion, we extended the revised manuscript with an extra section:

 “6. Role of Private Professionals” [page 4, lines 138 ff.]. 

However, as indicated in the title (Population-Based Linkage of Big Data in Dental Research), the main focus of the manuscript is on research (not dentistry in general). The wide field of dental medicine and healthcare of the dentist as routine work as well as the possible impact of social media in dentistry would be a standalone (new) publication. 

Thank you very much for the idea of Machine Learning. As an outlook, we included the idea of Machine Learning and Artificial Intelligence in the Conclusions [page 4, 152 ff.].